# Research Advances in Cadmium Uptake, Transport and Resistance in Rice (*Oryza sativa* L.)

**DOI:** 10.3390/cells11030569

**Published:** 2022-02-06

**Authors:** Jialiang Zhang, Yanchun Zhu, Lijuan Yu, Meng Yang, Xiao Zou, Changxi Yin, Yongjun Lin

**Affiliations:** 1National Key Laboratory of Crop Genetic Improvement, Huazhong Agricultural University, Wuhan 430070, China; zhangjialiang@webmail.hzau.edu.cn (J.Z.); zhuyanchun@webmail.hzau.edu.cn (Y.Z.); liulangx@webmail.hzau.edu.cn (M.Y.); zouxiao@webmail.hzau.edu.cn (X.Z.); yongjunlin@mail.hzau.edu.cn (Y.L.); 2MOA Key Laboratory of Crop Ecophysiology and Farming System in the Middle Reaches of the Yangtze River, College of Plant Science and Technology, Huazhong Agricultural University, Wuhan 430070, China; 3Agro-Products Processing Research Institute, Yunnan Academy of Agricultural Sciences, Kunming 650221, China; yulijuan1000@163.com; 4College of Life Science and Technology, Huazhong Agricultural University, Wuhan 430070, China

**Keywords:** cadmium, uptake, transport, Cd deposition, vacuolar sequestration, chelation, antioxidation, efflux

## Abstract

Rice (*Oryza sativa* L.) is one of the most important food crops, feeding half of the world’s population. However, rice production is affected by cadmium (Cd) toxicity. Due to an increase in Cd-contaminated soil and rice grains, and the serious harm to human health from Cd, research on Cd uptake, transport and resistance in rice has been widely conducted, and many important advances have been made. Rice plants absorb Cd mainly from soil through roots, which is mediated by Cd absorption-related transporters, including OsNramp5, OsNramp1, OsCd1, OsZIP3, OsHIR1,OsIRT1 and OsIRT2. Cd uptake is affected by soil’s environmental factors, such as the concentrations of Cd and some other ions in soil, soil properties, and other factors can affect the bioavailability of Cd in soil. Then, Cd is transported within rice plants mediated by OsZIP6, OsZIP7, OsLCD, OsHMA2, CAL1, OsCCX2, OsLCT1 and OsMTP1, from roots to shoots and from shoots to grains. To resist Cd toxicity, rice has evolved many resistance strategies, including the deposition of Cd in cell walls, vacuolar Cd sequestration, Cd chelation, antioxidation and Cd efflux. In addition, some unresolved scientific questions surrounding Cd uptake, transport and resistance in rice are proposed for further study.

## 1. Introduction

In recent years, the Cd content of farmland soils has continuously increased due to increasing discharges of industrial wastewater, waste gas and other residues, excessive use of metal-containing pesticides and fertilizers. The total concentration of Cd in agricultural soils is higher than that in non-agricultural soils [1]. Compared to survey results from 1990, the Cd content of Chinese soil has generally increased and continues to increase. About 20,000 hectares of farmland in China was contaminated with Cd [2]. The concentration of Cd in rice grains grown on farmland with heavy Cd contamination has exceeded the global safety standard level (≤0.4 mg/kg), rendering the grains inedible [3].

Rice is the main source of Cd in Chinese diet, and 55% of Cd intake by the Chinese population is from rice [4]. Cd enters the rice plant mainly through the roots and is transported to the shoots through sieve tubes and vessels and then transported to the grains. Cd enters the human body through the food chain, causing toxic effects on numerous systems, including the bones, heart and cerebral vessels and the nervous system. Moreover, Cd is not readily excreted from the human body and, therefore, causes long-term harm to human health. In addition, Cd pollution can affect rice growth and development and ultimately reduce rice yield [2].

Taken together, it is important to investigate the mechanisms of Cd uptake, transport and resistance in rice. In recent decades, scientists have made many important advances in understanding Cd uptake, transport and resistance in rice. Here, we introduce and discuss the research progress in this field. We also explore unresolved scientific questions regarding Cd uptake, transport and resistance in rice that demand further study in the future. The research progress in this field can be used to reduce the Cd-caused yield loss by enhancing rice resistance to Cd toxicity, produce safe rice grains by reducing Cd accumulation in rice grains and repair the Cd-contaminated paddy fields by cultivating and planting rice varieties with strong Cd enrichment.

## 2. Cd Uptake

Rice plants primarily absorb heavy metal ions through their roots. Roots are the first organ to contact Cd in the soil and also the first barrier used by rice to resist Cd toxicity. Cd concentration in rice roots is significantly higher than that in aerial organs [5]. Cd is actively absorbed by rice root mainly through the symplasmic pathway by which Cd is transported into the rice root cell by carrier protein by consuming energy; then, Cd is transported into the root vascular cylinder via plasmodesmata [6,7,8]. In addition, Cd is able to be passively absorbed by rice root through extracellular compartment, such as the intercellular space. If Cd concentration outside of the root cell is higher than that inside of the root cell, Cd can diffuse into the root cell without consuming energy [6,7,8].

### 2.1. Cd Uptake Mediated by Transporters

OsNramp5, a member of the rice natural resistance-associated macrophage protein (Nramp) family, is a membrane transporter involved in Cd and manganese (Mn) uptake by rice roots. This transporter is polarly localized on the distal sides of both exodermis and endodermis cells and is the main pathway through which rice roots absorb Cd and Mn (Figure 1, Table 1) [9]. Previous studies have demonstrated that the knockout or knockdown of OsNramp5 significantly reduces Cd uptake capacity of the root system, thereby reducing the Cd contents in rice shoots and grains [9,10]. A knockout of OsNramp5 significantly reduces the content of Cd in rice grains without affecting rice yield [5]. However, in Mn-deficient soils, OsNramp5-knockout rice mutants have reduced yield due to Mn deficiency, which is caused by a reduction in Mn uptake capacity, and translocation of Cd from roots to shoots was increased [5,9]. Interestingly, overexpression of OsNramp5 significantly increases the content of Cd in the roots, but markedly reduces it in aerial organs, such as straw and grains, by inhibiting the loading of Cd into the xylem [11]. This suggests that overexpression of OsNramp5 may be an effective strategy for reducing Cd accumulation in rice grains.

OsNramp1, a Cd transporter that can be up-regulated in response to Cd, participates in the absorption and transport of Cd in rice (Figure 1, Table 1) [12,13,14]. Overexpression of OsNramp1 increases the Cd concentration in rice leaves, whereas its knockout reduces Cd accumulation in rice shoots and grains. In addition, compared to wild-type yeast, yeast expressing rice OsNramp1 suffers more serious toxicity under Cd stress [13].

As the physical and chemical properties of Cd ions are similar to zinc (Zn) and iron (Fe) ions, Cd ions can be transported into rice cells through Zn and Fe transporters. The Zn and Fe transporters OsIRT1 and OsIRT2, which transport Zn and Fe across the membrane, have high affinities for Cd (Table 1). These two transporters are induced by Fe deficiency and can improve the resistance of rice plants to Fe deficiency, while also enhancing the absorption and transport of Cd and increasing the Cd content in rice (Figure 1) [15,16,17]. It has been reported that increased expression of OsIRT1 leads to increased Cd accumulation in rice [18]. In addition, NaCl can up-regulate the expression of IRT2 and promote the absorption of Cd by plants [19,20].

In addition, OsCd1, OsZIP3 and OsHIR1 are also involved in Cd uptake [21,22,23]. It has been reported that OsCd1 was localized on the plasma membrane of roots, and Cd accumulation in rice plants of OsCd1 knockout lines was significantly lower than that in wild-type rice (Figure 1, Table 1) [21]. Ectopic expression of OsZIP3 in yeast cells results in reduced absorption of Cd, leading to reduced Cd toxicity (Table 1) [22]. Ectopic expression of OsHIR1 in transgenic *Arabidopsis* reduces Cd accumulation in roots and buds, thereby enhancing Cd tolerance (Table 1) [23].

### 2.2. Cd Uptake Affected by Environmental Factors

Uptake of Cd by rice roots is influenced by environmental factors, such as the concentrations of Cd and some other ions in soil [49,50,51], soil properties, including soil pH [52], nitrogen source [53], soil organic matter [54], rhizosphere microorganisms [55], and other factors can affect the bioavailability of Cd in soil.

Generally, the uptake of Cd by rice is positively correlated with Cd concentration in soil [49]. It has been reported that some metallic elements, including Na, have enhancing effects on Cd uptake by plants [56,57]. By contrast, other metallic elements, such as calcium (Ca), magnesium (Mg), manganese (Mn), potassium (K) and silicon (Si) have inhibitory effects on Cd uptake [50,51,58]. In soil solution, excessive release of Mn through dissolution of Mn oxides inhibits the absorption of Cd by rice roots. Mn oxide with strong capacities of oxidation and adsorption can affect the bioavailability of Cd and the amount of Cd adsorbed onto other metal oxides. Moreover, Cd and Mn enter rice root cells through ion channels and carrier proteins, and Mn inhibits Cd uptake through competition with ion channels and carrier proteins [50].

Among soil properties, pH has a significant effect on the concentration of available Cd in the soil [52]. When the soil is highly acidic, the amount of available Cd in soil increases, promoting the absorption of Cd by rice roots. When the soil is highly alkaline, free Cd in the soil will become conjugated Cd, reducing the bioavailability of Cd and thus the uptake of Cd by rice [52]. It has been reported that excessive application of nitrogen fertilizer can lead to soil acidification and promote Cd uptake [53]. In recent years, soil acidification in China and other countries has been most often caused by excessive application of nitrogen fertilizer [53]. Soil acidification can reduce the adsorption of Cd by soil particles [59], increase the concentration of available Cd in soil and, therefore, increase the absorption and accumulation of Cd by rice [60,61]. By contrast, the application of alkaline substances, such as lime and grass ash, can improve the ability of soil to neutralize acids, enhance the Cd adsorption capacity of soil and reduce the bioavailability of Cd, thus reducing the absorption of Cd by rice roots and the accumulation of Cd in rice grains [59,62]. Additionally, salinity can increase the absorption of Cd by plant roots and reduce Cd resistance in *Raphanus sativus* L. [19,63], while high concentration of Cd has no effect on semi-halophyte *Mesembryanthemum crystallinum* L. [64]. Phenolic and carboxylic acid groups of soil organic matter can be complexed with Cd to form highly conjugated macromolecules, reducing the availability of Cd [54], while rhizosphere microorganisms can promote Cd uptake by rice roots through various mechanisms, including increasing the bioavailability of Cd in soils [55].

## 3. Cd Transport

### 3.1. Cd Transport from Roots to Shoots

Cd transport from roots to shoots is mediated mainly through the xylem, while only a small amount is transported to shoots through the phloem. Increasing evidence suggests that OsZIP6, OsZIP7, OsLCD, OsHMA2, CAL1 and OsMTP1 play important roles in mediating Cd transport in rice (Table 1) [24,25,26,27,28,29,30,31].

It has been reported that the ZIP family members, including OsZIP6 and OsZIP7, are involved in the transport of Cd in rice. OsZIP6 was predicted to be localized on the plasma membrane in root and shoot tissues and was involved in Cd transport from roots to shoots, but the specific mechanism by which OsZIP6 transports Cd in rice is still unclear [24]. In root, OsZIP7 plays a role in loading Cd into xylem (Figure 1). A knockout of OsZIP7 leads to the accumulation of large amounts of Cd in roots, decreasing Cd content in the grains [31].

Shimo et al. [25] screened the rice transfer DNA (T-DNA) mutant library for sequences related to Cd tolerance and obtained the Cd-resistant mutant *lcd*. The *lcd* seedlings grown on agar plates or in hydroponic culture exhibited increased tolerance of Cd toxicity. This study demonstrates that OsLCD is strongly expressed in root vascular bundles and OsLCD is involved in regulating the transport of Cd from the root to the shoot in rice. The knockout of OsLCD reduced the concentration of Cd in rice aerial organs and promoted rice growth under Cd stress. In addition, OsLCD is also expressed in phloem companion cells of leaves, suggesting the possibility that OsLCD is involved in regulating Cd transport from the leaf to the stem node through the phloem.

The transporter OsHMA2 is reported to mainly localize in the root pericycle cells of rice (Figure 1), and it plays a role in the xylem loading of Cd [24]. Compared to the wild type, the loss-of-function mutant of OsHMA2 shows significant inhibition of Cd transport from the root to the shoot [26,27]. In addition, the OsHMA2 gene is expressed in the phloem parenchyma cells of enlarged and diffuse vascular bundles of the stem node during the reproductive stage (Figure 2), which in turn increases the amount of Cd transported through the phloem. It has been confirmed that the concentration of Cd in OsHMA2-supressed rice leaves and grains is lower than that in wild type [26,27,28].

CAL1, a major quantitative trait locus gene that specifically regulates Cd accumulation in rice leaves, has been cloned. CAL1 is mainly localized to the root and the leaf sheath xylem parenchyma cells and can specifically chelate Cd with its three thiol groups, after which the Cd chelate is secreted into the xylem vessel and transported to the aerial vegetative organs, thereby increasing the Cd contents in rice straw and leaves (Figure 1) [29]. It has been reported that up-regulation of CAL1 can increase Cd accumulation in rice leaves. However, due to the tight binding of CAL1 to Cd, the transmembrane transport of Cd chelate into the phloem is prevented, and consequently, the transport of Cd from straw to grains is not promoted. Thus, up-regulation of CAL1 cannot increase Cd accumulation in rice grains [29].

OsMTP1, a transporter expressed in the root, the stem and the leaf, plays an important role in Cd transport [30]. Previous study demonstrated that when double-strand RNA interference (dsRNAi) plants of OsMTP1 were under Cd stress, the Cd content in the root was higher than that in the wild type, while Cd content in the shoot was lower than that in the wild type. These results suggest that down-regulation of OsMTP1 might impair Cd transport from roots to shoots. In addition, in the leaf, OsMTP1 is specifically expressed in sieve tube cells (Figure 2) [30], suggesting the possibility that OsMTP1 is involved in the mediation of Cd transport through the phloem in the leaf.

### 3.2. Cd Transport from Vegetative Organs to Grains

At the vegetative stage, the root-absorbed Cd is transported to the unelongated basal internodes mainly through the xylem and is then transported to the leaves through the xylem of the enlarged vascular bundle. During the early period of grain development, some of the Cd in leaves is reactivated and transported to stem nodes through the phloem and is then transported to grains through the phloem of the diffuse vascular bundle (Figure 2) [28,32,65,66]. Moreover, during the reproductive stage, some of the root-absorbed Cd is transported to the lower nodes through xylem and transported to the uppermost node through xylem and phloem; it is then transported to the grains through the phloem of the diffuse vascular bundles [28]. In addition, during the filling stage, a small amount of root-absorbed Cd is able to be transported to the stem and then directly transported to the grain through the xylem [32]. Once Cd is transported to the grain, it is fixed in the grain and is not further transported to other organs. At least 91% of Cd enters rice grains through the phloem [67], indicating that phloem transport plays an important role in Cd accumulation in rice grains. The above results show that Cd absorbed by roots is mainly transported to the aerial part through the xylem, while Cd enters grains mainly through the phloem, which indicates that most of the Cd needs to be transferred from the xylem to the phloem before being transported to grains.

It has been demonstrated that the stem node is an important organ that regulates the transfer of Cd from xylem to phloem, and the transfer process occurs in the enlarged and diffuse vascular bundles of the stem node [28,31,65]. In stem nodes, OsZIP7 is expressed in both the xylem and phloem parenchyma cells of the enlarged vascular bundle, which can facilitate the transfer of Cd from xylem to phloem (Figure 2, Table 1) [31]. Moreover, in stem nodes, Cd is able to be transferred between different vascular bundles, and Cd in the xylem of the enlarged vascular bundle is able to be transferred to the phloem of the diffuse vascular bundle [28,65]. Previous results demonstrated that three Cd transport genes, OsLCT1, OsZIP7 and OsHMA2, can express in stem nodes, and they may promote the transfer of Cd between vascular bundles in stem nodes. OsZIP7, which is expressed in xylem parenchyma cells of enlarged vascular bundles in stem nodes, is responsible for unloading Cd from the xylem of enlarged vascular bundles to the parenchyma cell bridge [31]. OsLCT1 can express in the parenchyma cell bridge (several layers of cells between the enlarged vascular bundles and the diffuse vascular bundles), and OsLCT1 may mediate the transport of Cd between the parenchyma cell bridge and the phloem parenchyma cells of diffuse vascular bundle [33]. In the diffuse vascular bundle of the stem node, OsHMA2 can load Cd from the phloem parenchyma cell to the phloem sieve [28]. These results indicate that OsLCT1, OsZIP7 and OsHMA2 may act synergistically to promote the transfer of Cd from enlarged to diffuse vascular bundles in stem nodes.

In addition, OsLCT1 has been described as a transporter that is highly expressed in the phloem plasma membrane of the vascular bundles in the uppermost node [33]. It has been reported that the expression level of OsLCT1 in the uppermost node at the grain maturity stage is about 100 times higher than that at the heading stage and that grain maturity is the key stage for Cd accumulation in rice grains [33]. The down-regulation of the OsLCT1 gene does not affect the transport of Cd in xylem but significantly inhibits its transport in phloem and decreases its content in rice grains. RNA interference of OsLCT1 expression has no negative effect on rice growth or the contents of other metals and mineral elements in rice grains. All this evidence suggests that, at the grain maturity stage, OsLCT1 is a crucial transporter for Cd transport from the uppermost node to rice grains (Figure 2, Table 1).

It has been reported that OsCCX2 can promote an upward transport of Cd in xylem (Figure 2, Table 1), and a knockout of the OsCCX2 gene reduces the transfer rate of Cd from roots to the aerial organs. OsCCX2 is mainly expressed in the xylem parenchyma cells of enlarged and diffuse vascular bundles in the stem node. OsCCX2 can load Cd into xylem vessels of diffuse vascular bundles and mediate Cd transport into grains through the xylem transport system of diffuse vascular bundles. The loss-of-function mutant of OsCCX2 significantly reduces the accumulation of Cd in grains [32]. In addition, OsCCX2 can also load Cd into xylem vessels of enlarged vascular bundles and mediate Cd transport into leaves through the xylem of enlarged vascular bundles.

Differences in Cd accumulation have been reported among rice subspecies. Previous results have demonstrated that OsCd1 is expressed in root cortex cells and root stele cells at the vegetative stage, and it can mediate the uptake of Cd by rice roots and may also participate in the transport of Cd from the root xylem to the aerial part, thus affecting the accumulation of Cd in the aerial organs of rice, including straw and grains [21]. The contents of Cd in the straw and grains of the loss-of-function mutant of OsCd1 are significantly decreased, and the growth and grain yield of the mutant are also severely reduced. Usually, the Cd content in *indica* rice grains is higher than that in *japonica* rice grains [68]. However, *indica* rice varieties carrying the *japonica* rice allele OsCd1^v449^ exhibit reduced Cd accumulation in grains, with almost no effect on growth or grain yield [21]. This evidence indicates that identifying and utilizing useful alleles of OsCd1 and other Cd accumulation-related genes may be an effective method to reduce Cd accumulation in rice grains.

## 4. Rice Resistance to Cd

To adapt to the Cd-contaminated soil environment, rice plants have evolved various strategies to resist Cd toxicity, including Cd deposition in cell wall, vacuolar sequestration, chelation, antioxidation and efflux [69,70,71].

### 4.1. Cd Deposition in the Cell Wall

The cell wall is the first defensive structure of plant cells to respond to adverse environmental conditions, such as Cd contamination [72]. After Cd enter the plant, some become attached to the cell wall, which prevents excessive Cd from reaching the cytoplasm, thereby protecting the cell from damage [72]. Root cell walls play an important role in this process, as the root is the organ most directly exposed to Cd in soil. In roots, Cd accumulation in the outer epidermal layer is usually more than that in cortical tissues [73]. When the concentration of Cd in the soil is low, most Cd taken up by plants is stored in the cell walls of roots [7]. Cd deposition in the cell wall will increase the content of Cd in rice roots, which in turn will decrease Cd content in the shoot by limiting the transport of Cd from the root to the shoot [72]. Similarly, increased Cd deposition in the cell walls of aerial vegetative organs will reduce the transport of Cd from these organs to the grains [34]. Previous result suggests that OsCDT1 may play an important role in increasing the resistance of rice to Cd toxicity (Table 1) [74]. OsCDT1 is reportedly targeted at both the cytoplasmic membranes and cell walls of plant cells, and OsCDT1 can chelate Cd on the cell surface, limiting Cd entry into the cell. Transgenic *Arabidopsis* plants overexpressing OsCDT1 accumulate less Cd than wild-type plants, and they display a Cd-tolerant phenotype [74].

After entering the root system, Cd is first fixed to the root cell wall by carbohydrates, which is the first barrier against Cd toxicity. The cell wall has a strong ability to adsorb Cd, which can prevent Cd from entering cells and improve the Cd stress resistance of plants. Pectin is an important Cd binding site in the cell wall, and rice cell walls with higher pectin contents adsorb Cd more quickly and exhibit greater Cd tolerance [75,76].

In contrast, a previous study revealed that heterologous overexpression of *Populus euphratica PeXTH* in tobacco reduced Cd accumulation in the cell and alleviated Cd toxicity by reducing the Cd binding sides on the cell wall [77]. It has been reported that xyloglucan endotransglucosylase/hydrolase (XTH) can alleviate Cd toxicity by decreasing Cd accumulation in plant root through the reduced Cd binding sites on the root cell wall [77]. Overexpression of *PeXTH* in tobacco promoted the degradation of xyloglucan, reduced the amount of xyloglucan in the cell wall and led to a decrease in Cd binding sites, thereby reducing Cd content of the cell wall, which in turn alleviated Cd toxicity in the transgenic tobacco plant [77].

### 4.2. Vacuolar Sequestration

In 2010, two research groups independently cloned OsHMA3, a gene that encodes a transporter (Table 1) [35,36]. OsHMA3 localizes on the vacuolar membrane and plays an important role in alleviating Cd toxicity in rice (Figure 1). OsHMA3 transports Cd from the cytoplasm to the vacuole, which in turn prevents transmembrane transport of Cd into the xylem. The down-regulation of OsHMA3 enhances Cd translocation to rice shoots, while its overexpression reduces Cd contents of the rice shoots and grains and alleviates Cd toxicity in OsHMA3-OE lines [35,36]. A loss-of-function mutation of the allele of OsHMA3 present in *japonica* rice varieties is the main reason for high accumulation of Cd in the seedlings and grains of some *japonica* varieties [37,78]. Other genes homologous to HMA3 may have the same function. For example, AtHMA3, one of the closest homologs of OsHMA3, can transport Cd and Zn in *Arabidopsis* and maintain Cd and Zn within the vacuoles [79]. In addition, in the Cd/Zn hyperaccumulator *Sedum plumbizincicola*, SpHMA3 plays a key role in mediating vacuolar Cd sequestration and Cd detoxification [80].

An ABC-type transporter, OsABCG43, may play a role in the sequestration of Cd into subcellular organelles, thereby reducing Cd toxicity (Table 1) [38]. In addition, the transport of Cd complex into vacuoles is another important detoxification pathway. The SpHMT1 transporter transports Cd complexes to vacuoles, and overexpression of SpHMT1 can significantly increase Cd tolerance in transgenic yeast [81]. In *Arabidopsis*, three ABCC-type transporters, AtABCC1, AtABCC2 and AtABCC3, have been reported to improve tolerance of Cd by transferring phytochelatin (PC)–Cd chelates from the cytoplasm to the vacuoles, and a knockout of these genes causes a hypersensitive response of *Arabidopsis* to Cd [82,83]. AtABCC1 and AtABCC2 can also increase the tolerance of *Arabidopsis* to mercury (Hg) by transferring PC–Hg chelates from the cytoplasm to the vacuoles [82]. However, the transporters that can transport PC–Cd and PC–Hg chelates into vacuoles have not yet been identified in rice.

### 4.3. Chelation

The chelation of Cd plays an important role in plant tolerance of Cd toxicity. Chelating agents in plants can prevent contact between Cd and organelles through Cd chelation and impede biochemical reactions between Cd and other substances, thereby reducing Cd toxicity [39,84,85,86].

Glutathione (GSH) is a tripeptide that can form an active transport complex of bis (glutathionato) cadmium (Cd·GS_2_) (Figure 1). Yeast cadmium factor 1 (YCF1) is able to transport Cd·GS_2_ into the vacuoles, thus reducing the toxicity of Cd to plants [85]. PC is a Cd-binding peptide that is synthesized using GSH as substrate and catalyzed by PC synthetase. Cysteine synthesis is the rate-limiting step of PC synthesis, and transgenic plants expressing the rice cysteine synthase gene OsRCS1 have increased Cd tolerance compared to wild-type plants (Table 1) [40]. Moreover, overexpression of the rice PC synthesis genes OsPCS5 and OsPCS15 in *ycf1*, a loss-of-function mutant of *YCF1*, can enhance the resistance of *ycf1* to Cd toxicity (Table 1) [41].

Binding peptides can bind 90% of the Cd taken up by plants, providing resistance against Cd toxicity [84]. The *Arabidopsis* mutant *cad1*, which lacks PC synthase activity, is hypersensitive to Cd [87]. PC can chelate Cd, thus reducing damage to the photosynthetic system of *Brassica juncea* caused by Cd [88]. Overexpression of the *Arabidopsis* PC synthase gene AtPCS in rice significantly enhances tolerance of Cd [89]. In addition, the expression of the wheat PC synthase gene TaPCS1 in tobacco and the *Arabidopsis* PC synthase gene AtPCS1 in *Brassica juncea* increases PC synthesis and enhances the tolerance of both transgenic plants of Cd [90,91]. High salinity and Cd stresses can synergistically up-regulate the expression of PCS1 in semi-halophyte *Mesembryanthemum crystallinum* and promote PC synthesis and enhance the resistance of halophytes to Cd under high salinity [19,20]. In contrast, overexpression of OsPCS5 and OsPCS15 in transgenic *Arabidopsis* results in a hypersensitive response to Cd and reduces resistance to Cd [41]. Similarly, overexpression of AtPCS1 in tobacco and TaPCS1 in rice also leads to hypersensitive responses of transgenic plants to Cd, along with reduced resistance to Cd [92,93]. Taken together, these findings show that overexpression of the same PC synthase gene in different plant species may lead to different degrees of Cd resistance, and overexpression of different PC synthase genes in the same plant species may also lead to differences in Cd resistance. Possible explanations for these differences among plant species and PC synthase genes include the cells of different plant species having different microenvironments, the enzymes produced due to overexpression of PC synthase genes exhibiting different enzyme activities and ectopic expression levels of different PC synthase genes being different, even within a plant species [92]. In addition, although PC content is closely correlated with the accumulation of Cd in vivo [89,93,94], the threshold level of Cd accumulation in vacuoles may differ among species, which may lead to differences in Cd resistance.

Plant roots can secrete citric acid into the soil, which can increase the availability of Cd in soil and thus promote the absorption of Cd by plants. On the other hand, citric acid can chelate free Cd ions and reduce their toxic effects on plants, as well as enhance the activities of antioxidant enzymes and alleviate the lipid peroxidation induced by Cd stress [86,95,96]. In addition, some organic compounds such as humic acid will chelate with Cd in the process of vascular transport of Cd, thus affecting the bioavailability of Cd [97,98].

OsHsfA4a can induce the expression of the *metallothionein* (*MT*) gene and thus enhance the tolerance of rice to Cd via the chelation between MT and Cd (Table 1) [39]. The expression of the rice rgMT gene in *Escherichia coli* (*E. coli*) enhances the Cd resistance of transgenic *E*. *coli* (Table 1) [42].

### 4.4. Antioxidation

When rice is exposed to Cd stress, it produces excessive quantities of reactive oxygen species (ROS), such as superoxide anion radicals, hydrogen peroxide and hydroxyl radicals. Excessive ROS disrupt the redox balance of rice plants, leading to a dysfunction of the antioxidant system, which in turn causes membrane lipid peroxidation and other types of oxidative damage, including programed cell death [99].

GSH plays important roles during the antioxidation process of rice exposed to Cd. It has been reported that glutathione synthetase (GS) is an important enzyme involved in catalyzing GSH biosynthesis, and the overexpression of the gene encoding γ-glutamylcysteine synthetase (γ-ECS) and the *gshII* gene encoding GS from *E*. *coli* in *Brassica juncea* promotes glutathione synthesis and enhances Cd tolerance of transgenic *Brassica juncea* [100,101]. Previous studies demonstrated that overexpression of the γ-ECS can also strengthen *Arabidopsis* resistance to arsenic [102]. GSH can remove excessive ROS and bind Cd, effectively reducing the toxic effects of Cd on rice [103]. Moreover, GSH can enter the nucleus during cell division and proliferation to regulate the cell cycle, and it can also remove ROS directly or indirectly, thus playing an important role in maintaining the redox homeostasis of cells [104]. In addition, GSH can inhibit lipid peroxidation through the GSH peroxidase and GSH reductase system, thereby protecting cell membrane, and it can directly reduce lipid free radicals and lipid peroxide free radicals to block the chain reaction of lipid peroxidation [105]. In addition, salinity stress can enhance the activities of antioxidative enzymes catalase and GSH peroxidase to alleviate the oxidative stress caused by Cd toxicity [106].

Previous report demonstrated that OsCLT1, which is localized to plastids, can regulate the Cd tolerance of rice plants by alleviating the oxidative stress via the promotion of GSH biosynthesis (Table 1) [43]. The loss-of-function mutation of OsCLT1 led to decreased GSH and PC concentrations in root cells of mutant rice, resulting in hypersensitivity to Cd [43]. Agrawal et al. [44] suggested that ROS induced by Cd toxicity, particularly H_2_O_2_, regulates antioxidant activity in rice plants by inducing the expression of OsMSRMK3 and OsWJUMK1 (Table 1).

It has been reported that jasmonic acid, jasmonoyl–isoleucine, methyl jasmonate and 12-oxygen phthalate acid (a jasmonic acid precursor) are signal molecules involved in the regulation of rice growth and stress responses [107]. Methyl jasmonate can reduce the absorption of Cd and increase the activities of antioxidant enzymes, such as peroxidase, superoxide dismutase, catalase and glutathione reductase, which in turn reduces excessive hydrogen peroxide and superoxide anion free radicals, thereby reducing oxidative damage caused by Cd to rice seedlings [108]. In addition, the OsAUX1 gene can regulate the growth of primary roots and root hairs and improve the capacity to resist oxidative stress under Cd exposure (Table 1), and thus a loss-of-function mutant of OsAUX1 in rice exhibits severe oxidative damage due to Cd exposure [45].

### 4.5. Efflux

OsHMA9, which encodes a heavy metal efflux transporter that is located in the plasma membrane, has Cd, Cu, Zn and Pb efflux activities (Figure 1, Table 1). A knockout of the OsHMA9 gene results in greater accumulation of Cd in mutant plants compared to wild-type plants, which in turn leads to a hypersensitive response of the mutant to Cd [46]. Moreover, plant ZIP proteins (zinc and iron transporters), a family of metal transporters, are involved in the absorption and distribution of Zn and Cd. OsZIP1, a transporter that is widely expressed in rice roots (Table 1), plays an important role in metal detoxification and prevents the excessive accumulation of Zn, Cu and Cd in rice. When Zn, Cu and Cd levels in the environment are excessive, OsZIP1 acts as a metal efflux transporter from rice cells [47]. In addition, OsABCG36/OsPDR9, a G-type ATP-binding transporter, is involved in Cd resistance in rice (Table 1) [48]. It has been confirmed that OsABCG36 is localized to the plasma membrane by using a transient expression system in rice protoplast, and Cd could significantly induce the expression of OsABCG36 in mature root zones, as well as root tips. Under high Cd stress conditions, compared to the wild type, the accumulation of Cd in the root cell sap of OsABCG36 knockout rice was significantly elevated, as was sensitivity to Cd, but the accumulation of Cd in aerial organs was essentially unchanged. This pattern indicated that OsABCG36 is not involved in the accumulation of Cd in aerial organs, but instead enhances the resistance of rice to Cd by promoting the efflux of Cd or Cd conjugates from root cells [48].

## 5. Conclusions and Perspectives

Cd pollution of farmland is a serious problem. In recent decades, scientists have made many important advances in understanding Cd uptake, transport and resistance in rice, but some of the molecular mechanisms underlying these processes are still unknown. Previous reports indicated that the expression of OsCd1 in the root cortex may promote the absorption of Cd by rice roots, but it is unclear whether the expression of OsCd1 in the root stele can facilitate the transport of Cd from the root to the shoot by promoting Cd loading. OsCDT1 is located in the cytoplasmic membrane and can restrict the entry of Cd into cells by chelating Cd on the cell surface, thereby enhancing the tolerance of plants to Cd. However, the chelation site for Cd on OsCDT1 has not yet been reported, and the chelation mechanism has not been well studied. Although PC plays an important role in the process of Cd chelation and the mitigating of Cd toxicity, overexpression of PC synthase genes leads to increased Cd tolerance in some plants and decreased Cd tolerance in others, and the different mechanisms for these opposite effects need further confirmation. Previous studies have shown that loss-of-function mutation of OsAUX1 results in increased Cd sensitivity due to the severe oxidative damage in the rice mutant under Cd exposure, but the regulatory mechanism of OsAUX1 in the antioxidation process under Cd exposure remains unclear. OsABCG36 promotes Cd efflux from rice roots, thereby improving resistance to Cd. However, the form (ionic or chelated) of Cd that is excreted due to OsABCG36 remains unknown, and the molecular mechanism through which OsABCG36 promotes Cd efflux has not yet been elucidated.

The Cd content in *indica* rice grains is generally higher than that in *japonica* rice grains [71], and *indica* rice varieties that carry the *japonica* rice allele OsCd1^v449^ exhibit reduced Cd accumulation in grains [22]. Loss-of-function mutation of the OsHMA3 alleles in *japonica* rice varieties is the main reason for the accumulation of Cd in the grains of some *japonica* rice varieties [81,82]. However, whether natural variation in other genes also contributes to the differences in Cd content between the *indica* and *japonica* rice varieties remains obscure. Moreover, the transport of Cd and Cd complexes into vacuoles is also an important detoxification method. However, the transporters that can transport Cd complexes such as PC–Cd and Cd·GS_2_ into the vacuoles of rice cells have not been identified. In future research, the unresolved scientific questions concerning Cd uptake, transport and resistance in rice may be revealed through the construction of mutants and use of molecular, genetic, physiological and multi-omics techniques, and bioinformatics. In addition, it is of great significance to identify unknown genes, which are related to Cd uptake, transport and resistance in rice, and to investigate the functional mechanisms of these genes. Additionally, with the intensification of climate change, the decomposition of soil organic matter is also strengthening [109], which may lead to an increase in Cd bioavailability in soil and may increase Cd toxicity to rice. The underlying mechanism by which climate change affects Cd bioavailability in soil needs further investigation. The research progress in this field can be used to reduce the Cd-caused yield loss by enhancing rice resistance to Cd toxicity, produce safe rice grains by reducing Cd accumulation in rice grains and repair the Cd-contaminated paddy fields by cultivating and planting rice varieties with strong Cd enrichment.

## Figures and Tables

**Figure 1 cells-11-00569-f001:**
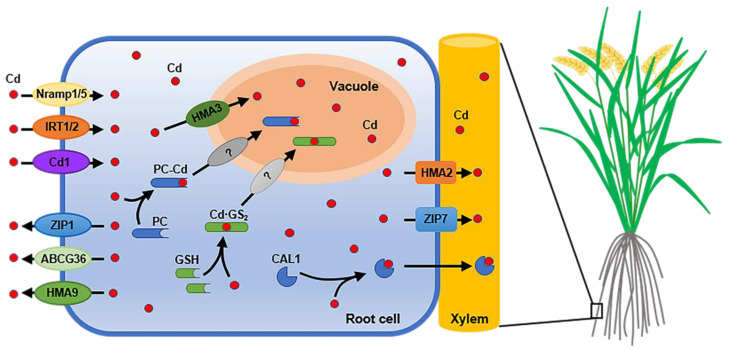
Schematic diagram of Cd uptake, efflux, vacuolar sequestration and loading into xylem in rice root. Cd is absorbed by root cells from soil, which is mediated by OsNramp1, OsNramp5, OsIRT1, OsIRT2 and OsCd1. After entering root cells, Cd is able to be transported into vacuole by OsHMA3 for sequestration. Cd is also to be chelated with PC or GSH and then transported into vacuole in the form of chelates, but the transporters that can transport PC–Cd and Cd·GS_2_ into vacuole have not been identified. OsZIP1, OsHMA9 and OsABCG36 mediate the efflux of Cd, while OsHMA2, OsZIP7 and CAL1 mediate the loading process of Cd into xylem in rice root.

**Figure 2 cells-11-00569-f002:**
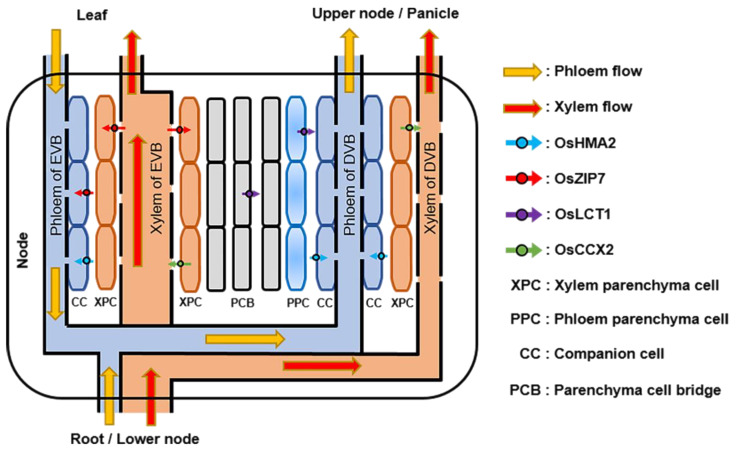
Schematic diagram of Cd transport in rice nodes. OsHMA2, localized in the phloem parenchyma and companion cells of both enlarged and diffuse vascular bundles in stem nodes, mediates the loading of Cd into the phloem for the preferential distribution to the upper nodes and panicle. OsZIP7, localized in both xylem and phloem parenchyma cells of the enlarged vascular bundles, can transport the Cd from xylem to phloem and unload Cd from xylem to parenchyma cell bridge. OsLCT1, localized in parenchyma cell bridge and phloem parenchyma cells of diffuse vascular bundles, mediates the intervascular transfer of Cd and promotes the transport of Cd to the grain through phloem. OsCCX2, localized in the xylem parenchyma cells of enlarged and diffuse vascular bundles, mediates the loading of Cd into the xylem and promotes Cd transport into grains.

**Table 1 cells-11-00569-t001:** Cd uptake, transport and resistance-related genes in rice.

Gene	GenBank	Protein Product	Function	References
*OsNramp5*	LOC_Os07g15370	Natural resistance-associated macrophage protein	Cd uptake	[5,9,10,11]
*OsNramp1*	LOC_Os07g15460	Natural resistance-associated macrophage protein	Cd uptake	[12,13,14]
*OsIRT1*	LOC_Os03g46470	Iron-regulated transporter	Cd uptake	[15,16,17,18]
*OsIRT2*	LOC_Os03g46454	Iron-regulated transporter	Cd uptake	[16,17,19,20]
*OsCd1*	LOC_Os03g02380	Major facilitator superfamily	Cd uptake	[21]
*OsZIP3*	LOC_Os04g52310	Zinc- and iron-regulated transporter	Cd uptake	[22]
*OsHIR1*	LOC_Os08g30790	Hypersensitive induced reaction protein	Cd uptake	[23]
*OsZIP6*	LOC_Os05g07210	Zinc- and iron-regulated transporter	Cd transport	[24]
*OsLCD*	LOC_Os01g72670	Low cadmium	Cd transport	[25]
*OsHMA2*	LOC_Os06g48720	P-type heavy metal ATPase	Cd transport	[26,27,28]
*CAL1*	LOC_Os02g41904	Defensin-like protein	Cd chelationCd transport	[29]
*OsMTP1*	LOC_Os05g03780	Metal tolerance protein	Cd transport	[30]
*OsZIP7*	LOC_Os05g10940	Zinc- and iron-regulated transporter	Cd transport	[31]
*OsCCX2*	LOC_Os03g45370	Cation/calcium (Ca) exchanger	Cd transport	[32]
*OsLCT1*	LOC_Os06g38120	Low affinity cation transporter	Cd transport	[33]
*OsCDT1*	LOC_Os03g45370	Cation/calcium (Ca) exchanger	Cd deposition	[34]
*OsHMA3*	LOC_Os07g12900	P-type heavy metal ATPase	Cd sequestration	[35,36,37]
*OsABCG43*	LOC_Os07g33780	ATP binding cassette (ABC)-type transporter	Cd sequestration	[38]
*OsHsfA4a*	LOC_Os01g54550	Heat shock transcription factor gene	Cd chelation	[39]
*OsRCS1*	LOC_Os12g42980	Cysteine synthase	Cd chelation	[40]
*OsPCS5*	LOC_Os06g01260	Phytochelatin synthase	Cd chelation	[41]
*OsPCS15*	LOC_Os05g34290	Phytochelatin synthase	Cd chelation	[41]
*rgMT*	LOC_Os11g47809	Type 1 metallothionein	Cd chelation	[42]
*OsCLT1*	LOC_Os01g72570	CRT-like transporter	Antioxidation	[43]
*OsMSRMK3*	LOC_Os06g48590	Mitogen-activated protein kinase	Antioxidation	[44]
*OsWJUMK1*	LOC_Os01g47530	Mitogen-activated protein kinase	Antioxidation	[44]
*OsAUX1*	LOC_Os01g63770	Auxin transport protein	Antioxidation	[45]
*OsHMA9*	LOC_Os06g45500	P-type heavy metal ATPase	Cd efflux	[46]
*OsZIP1*	LOC_Os01g74110	Zinc- and iron-regulated transporter	Cd efflux	[47]
*OsABCG36*	LOC_Os01g42380	PDR-type ABC transporter 9	Cd efflux	[48]

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
