# Peer review of "Research Advances in Cadmium Uptake, Transport and Resistance in Rice (Oryza sativa L.)"

_cells, 2022, doi:10.3390/cells11030569_

Round 1
Reviewer 1 Report
Comments to MS „Research advances in cadmium uptake, transport, and resistance in rice (Oryza sativa L.) ” written by Jialiang Zhang et al.
It was demonstrated many times that heavy metal stress and resulting level of reactive oxygen species play an important regulatory role in aclimatory and defence mechanisms in all plants. Some heavy metals can accumulate in different plant organs, however, our knowledge about such response, physiological meaning and regulation of dedicated gene in different plant tissues during application of Cd is rather scant. The general intention of this review MS is good and the problem is well stated. Results presented in this MS have big scientific potential and there are several novel aspects resulting from this what has previously been published in the literature. This manuscript focuses on proteins participating in heavy metal transportation and detoxification. All stated details dealing with the structure and evolution of these proteins need interpretation. We know also that these genes play important role in plants exposed to other stresses. However, there are some issues the authors should address before this manuscript could be considered acceptable.
Key words: It is not necessary to repeat expressions used in the title.
Abstract: Some statements resulting from here reported papers should be compared with results obtained in experiments on other resistant and sensitive plants. Several sentences on this can help to understand specificity of this what was obtained in experiments on rice.
Introduction:
Potential reader of this MS will expect to understand the meaning of all stated facts on plant resistance to Cd in relation to other plants. Recently (2019-2021) several papers were published (Plants and Journal of Plant Physiology) describing Cd-caused effects. Expression of many genes depend on physiological plant status and plant tissue and also their ability to survive in high salinity (glicophytes versus halophytes). This can be discussed when compared to other plants much more resistant than rice. This would be also important when discussing Cd trafficking to grains. Despite of this I would advise to mention with more details resistance to other toxic heavy metals.
Major criticism:
It is well known that changes in activity of metal transporting proteins are crucial for plant resistance to cadmium. Their role in plant reaction also to other stresses need to be elucidated. In this MS we learn a lot about the location and structure of these proteins. We know that some similar proteins play a role in transportation of other heavy metals and together with antioxidative enzymes are responsible for cross tolerance mechanism. They participate in plant adaptation to other stresses. This would help to understand plant reaction in given circumstances.
This MS should also point out the novel aspects of presented results for physiology of plants. Authors of this paper should not present possible increase in expression of these proteins only in terms of stress, as sometimes these proteins are important in gaining other micronutrients.
Generally, the paper is concisely written. After correction this can be very nice contribution to MDPI-Cells
Reviewer 2 Report
The manuscript is well constructed, but it needs to be improved in several segments as well in English grammar and style intensively. The author needs to confirm and clarify the novelty of their review. There is plenty of similar studies related to Cd uptake and routes in crops like rice. In addition, some aspects of text formatting are confusing and the work objective is not well established throughout the text, while some sections are too short. For instance, the authors from the beginning of the manuscript force term “Cd toxicity”. However, even at much lower concentrations (below the phyto-toxic threshold), the problem with Cd is in its “contamination” not only “toxicity”. Thus, it needs to be clarified given Cd toxicity in plants is induced under relatively higher dosages, while the problem with Cd contaminations starts at much lower levels. For this you can consult and refer some of the recent related publications (check and their reference lists as well):
https://doi.org/10.1016/j.scitotenv.2019.134887
https://doi.org/10.1016/j.ecoenv.2019.01.021
https://doi.org/10.3390/ijerph16030373
Next, the authors have addressed several critical points in Cd metabolism like in:
Section 2.2. Cd uptake affected by environmental factors.
Section 3.1. Cd transport from roots to shoots
However, this section needs to be additionally extended by explaining some other crucial factors in Cd biogeochemistry, uptake and metabolism like salinity and alkalinity, as some of the widely present constraints across the paddy agroecosystems. To accomplish this task authors can consult/refer to some of the recent studies listed above (and their references lists) related to this problem
In the section Cd transport from roots to shoots, there is no information about Cd chemical speciation (e.g. organic and inorganic complexes) in the plant vascular tissues, which needs to be elaborated with several examples.
Finally, the last section, conclusion and future perspectives need to be extended with addressing some of the problems related to relevant Climate change adaptation.
The last part in the last statement .... "and repair the Cd-contaminated paddy fields by cultivating and planting rice varieties with strong Cd enrichment", are you sure about that?
Please, reconsider it.
Round 2
Reviewer 1 Report
Thank you for the extensive answers to my comments. I think that now the manuscript is much improved
Reviewer 2 Report
The authors have answered appropriately all addressed remarks and comments, however some issues still exist related to English grammar and style.